# Hepatitis C (HCV) Reinfection and Risk Factors among Clients of a Low-Threshold Primary Healthcare Service for People Who Inject Drugs in Sydney, Australia

**DOI:** 10.3390/v16060957

**Published:** 2024-06-13

**Authors:** Phillip Read, Bruce Zi Huan Tang, Edmund Silins, Anna Doab, Vincent J. Cornelisse, Rosie Gilliver

**Affiliations:** 1Kirketon Road Centre, Kings Cross, P.O. Box 22, Sydney, NSW 1340, Australia; edmund.silins@health.nsw.gov.au (E.S.); anna.doab@health.nsw.gov.au (A.D.); rosie.gilliver@health.nsw.gov.au (R.G.); 2Kirby Institute, UNSW Australia, Sydney, NSW 2032, Australia; 3UNSW Medicine and Health, Sydney, NSW 2052, Australia; bruce.tang@student.unsw.edu.au; 4National Drug and Alcohol Research Centre (NDARC), UNSW Australia, Sydney, NSW 2052, Australia

**Keywords:** hepatitis C, reinfection, risk factors, vulnerable populations, primary healthcare

## Abstract

Hepatitis C (HCV) reinfection studies have not focused on primary healthcare services in Australia, where priority populations including people who inject drugs (PWID) typically engage in healthcare. We aimed to describe the incidence of HCV reinfection and associated risk factors in a cohort of people most at risk of reinfection in a real-world community setting. We conducted a secondary analysis of routinely collected HCV testing and treatment data from treatment episodes initiated with direct-acting antiviral (DAA) therapy between October 2015 and June 2021. The overall proportion of clients (N = 413) reinfected was 9% (N = 37), and the overall incidence rate of HCV reinfection was 9.5/100PY (95% CI: 6.3–14.3). Reinfection incidence rates varied by sub-group and were highest for Aboriginal and/or Torres Strait Islander people (20.4/100PY; 95% CI: 12.1–34.4). Among PWID (N= 321), only Aboriginality was significantly associated with reinfection (AOR: 2.73, 95% CI: 1.33–5.60, *p* = 0.006). High rates of HCV reinfection in populations with multiple vulnerabilities and continued drug use, especially among Aboriginal and Torres Strait Islander people, highlight the need for ongoing regular HCV testing and retreatment in order to achieve HCV elimination. A priority is resourcing testing and treatment for Aboriginal and/or Torres Strait Islander people. Our findings support the need for novel and holistic healthcare strategies for PWID and the upscaling of Indigenous cultural approaches and interventions.

## 1. Introduction

Hepatitis C virus (HCV) remains a major public health problem, with over 58 million infections worldwide and 1.5 million new cases each year [1]. Consequently, it is a leading cause of hepatocellular carcinoma and cirrhosis [1]. The World Health Organization (WHO) aims to reduce HCV incidence by 80% and achieve a 65% reduction in mortality from viral hepatitis by 2030 [1,2].

Direct-acting antiviral therapies (DAAs) for HCV have proven to have a treatment efficacy of >95% cure [3,4,5] and excellent tolerability [3,6], and they have dramatically increased treatment access and completion compared to interferon-based therapies [3]. These DAA medications have permitted broader access and prescriber bases, with outcomes comparable between primary care and tertiary settings [3]. The current cost of DAA treatment is still high for nearly all countries [1,7,8].

Neither treatment nor spontaneous clearance confer immunity against subsequent HCV infection [9], which means reinfection is a possibility. Studies in high-income countries have identified injecting drug use and HIV co-infection [3,9,10,11] as major risk factors for HCV reinfection. Other risk factors include incarceration and condomless sex among men who have sex with men (MSM) [3,12]. A recent meta-analysis assessed post-treatment HCV reinfection rates in people who inject drugs (PWID) and found a rate of reinfection following successful HCV treatment of 5.9 per 100 person years (PY) (95% CI: 4.1–8.5) among people with recent drug use (injecting or non-injecting), 6.2/100PY (95% CI: 4.3–9.0) among people with recent injecting drug use, and 3.8/100/PY (95% CI: 2.5–5.8) among those receiving opioid agonist therapy (OAT). Reinfection rates were lowest among people receiving OAT with no recent drug use (1.4/100PY, 95% CI: 0.8–2.6), compared to people with recent drug use, and similar following either interferon-based or direct-acting antiviral based therapy [13].

In Australia, in 2015, an estimated 180,000 people were living with untreated HCV, with an estimated 80% of all infections diagnosed [14]. Since the inclusion of DAAs in Australia’s Pharmaceutical Benefits Scheme (PBS) in 2016, over 100,000 individuals have been prescribed treatment for HCV infection [2,14,15,16]. Australia was one of the first countries to introduce universal treatment of HCV without restriction, including treatment of reinfection, whilst also having one of the lowest costs for DAA treatment out of all high-income countries [1,16]. While over 50% of people living with HCV in Australia have been successfully treated [17,18], DAA uptake has declined (from 11,310 in 2019 to 8100 in 2020 [2]), and an estimated 14,000 annual treatments are required nationally to reach WHO elimination goals [19].

If we are to achieve HCV elimination, it is increasingly important to understand the role and drivers of HCV reinfection to help focus prevention efforts and effectively utilise large investments in HCV treatment. This is particularly important among people at increased risk of HCV infection, for example, PWID, as these populations may be reluctant to engage with mainstream healthcare services that either have financial or access barriers and are not oriented to this population due to fear of stigma and discrimination [17,20,21,22]. Ongoing harm reduction such as high-coverage needle and syringe programs and access to opioid agonist therapy is also an essential component of reinfection prevention.

Australia is in an ideal position to conduct studies of community-based treatment for HCV, as 38% of DAA prescriptions are from non-specialist community prescribers [3]. However, few studies have described the characteristics of clients treated for HCV and followed up for reinfection in primary care in Australia [23].

In this study, we examined HCV reinfection in a cohort of people most at risk of reinfection in a real-world, single-service, community setting. We aimed to accomplish the following:Identify the demographic, behavioural, and clinical risk factors associated with being diagnosed with HCV reinfection at our service among PWID;Adjust the associations reported to identify independent risk factors for reinfection;Determine the incidence of HCV reinfection diagnosis overall and by sub-group.

## 2. Materials and Methods

### 2.1. Setting

The Kirketon Road Centre (KRC) is a publicly funded primary healthcare service with a harm reduction ethos located in inner Sydney, Australia, focused on the prevention, treatment, and care of viral hepatitis, HIV, and sexually transmissible infections among marginalised populations including PWID. KRC offers walk-in, free, non-judgemental health care at fixed sites and outreach locations, with the option of registering with a pseudonym if required. KRC operates an integrated needle and syringe program staffed by peer workers and health educators, provides opioid agonist therapy in-house, and is located adjacent to the Sydney Medically Supervised Injecting Centre.

### 2.2. Design, Timeframe, and Participants

We conducted an observational study of clients treated for HCV at KRC between October 2015 (when DAAs first became available at KRC through expanded access programs prior to government listing in March 2016) to June 2021. This was an analysis of routinely collected HCV testing, treatment, and reinfection data. All clients treated with DAAs by our service during this period were included in the analysis. We defined participants as having HCV infection any time they had a positive HCV RNA PCR test.

### 2.3. Data Sources

We retrospectively generated a single HCV dataset from a treatment database used to track treatment initiations and outcomes, routine electronic medical records, and pathology results.

Clinicians had entered clinical information into the HCV treatment database at initial assessment, treatment, and sustained virologic response (SVR) 12 weeks after stopping DAAs (SVR12). The database included demographic characteristics, risk factors for HCV infection, SVR12 test results, subsequent HCV PCR test dates and results, and loss to follow-up. For the purposes of this study, loss to follow-up was defined as having no record of further RNA testing after treatment initiation. Where data were missing from the treatment database, we manually searched electronic medical records and pathology results. Repeat testing data were included until June 2022.

### 2.4. Variables

The analyses included the following variables.

#### 2.4.1. Demographic, Behavioural, and Clinical Risk Factors

Data included in the analysis were based on the most recent available at the time of treatment.

Variables collected included age, in years; self-reported gender (male/female/non-binary (including other); Aboriginal and/or Torres Strait Islander identity; Australian-born; ever incarcerated including juvenile detention; whether someone was a man who has sex with men (cis or trans) (MSM); homelessness in the prior 12 months; recent (6 month) injecting drug use; current receipt of opioid agonist therapy; HIV status; hepatitis B surface antigen positivity (sAg); or the presence of cirrhosis (defined as Fibroscan > 12.5 kpa).

#### 2.4.2. HCV Reinfection

HCV reinfection was defined and assessed in two ways:*Proportion ever diagnosed with HCV reinfection.* The ever-reinfected group included all individuals identified as having new HCV infection (RNA positivity) after a confirmed cure regardless of whether exact testing dates were documented by us (e.g., we determined they were reinfected as they returned to our service on a second treatment course, or reinfection was detailed in a referral letter without exact testing dates). Individuals who did not fit this criterion were classified as not reinfected (i.e., individuals in whom we did not detect reinfection or did not test for reinfection were included as not reinfected). This definition was used to compute the proportion diagnosed as reinfected and the associations between independent variables and HCV reinfection in bivariate and multivariable models.*HCV reinfection incidence based on interval HCV RNA testing data*. This definition was based on the last undetectable and the first detectable HCV test date. Treatment episodes or test results where an undetectable HCV test result was followed by a detectable HCV test result after the original SVR12 due date were classified as reinfections. People with interval negative testing data were classified as not reinfected.

This second definition excluded treatment episodes where HCV reinfection was documented but for which the exact dates of first confirmatory RNA tests were not available.

Clients who did not receive HCV testing at the time of their SVR12 due date but received a negative HCV RNA result after this date were assumed to be cured at the time of SVR12 unless they had been retreated.

Time at risk of reinfection began at a client’s SVR12 due date and ended at, (1) for non-reinfected clients, the date of their last negative HCV result post SVR12 and, (2) for clients who were reinfected, the date halfway between their last HCV RNA-negative result and their earliest HCV RNA-positive result.

Clients without HCV RNA test data post-treatment were excluded from the incidence rate calculation as their time at risk of reinfection could not be calculated based on available data.

### 2.5. Statistical Analysis

There were five components to the main analysis:


*Among clients with recent injecting drug use (6 months)*
The demographic, behavioural, and clinical characteristics (based on the most recent data available) of DAA-treated clients with recent injecting drug use were described and the proportion of reinfected reported.Bivariate associations between independent variables (i.e., demographic, behavioural, and clinical characteristics) and HCV reinfection were examined using logistic regression models to identify factors associated with reinfection (*p* < 0.05).Variables that were statistically significant at the *p* < 0.15 level in the bivariate analysis were included in a multivariable logistic regression model to identify independent risk factors for reinfection.To investigate the extent to which loss to follow-up may have biased results, we used logistic regression models and examined the bivariate association between client characteristics and being lost to follow-up.



*Among clients with interval HCV RNA testing data*
5.An incidence rate of reinfection per 100PY of follow-up was generated for each independent variable (using the definition of reinfection based on interval testing data). This was calculated for all DAA-treated clients with interval testing data and a subset of people with recent (6 month) injecting drug use.


Continuous variables were tested for normality using the Shapiro–Wilk W test for normal data; means (and standard deviations) were reported if normally distributed and medians (and range) reported if non-normally distributed. Proportions (n/N) were reported for categorical variables. Odds ratios, 95% confidence intervals, and *p*-values were reported. The reinfection rate per 100 person years was calculated. Statistical significance was defined by *p* < 0.05. All analyses were performed on STATA v18.0 [24].


*Aboriginal and/or Torres Strait Islander interpretation:*


An Aboriginal and/or Torres Strait Islander reference group was established comprising KRC clients, KRC Aboriginal staff, and local Aboriginal and/or Torres Strait Islander community members to provide recommendations and advice on interpreting outcomes relating to Aboriginality.

## 3. Results

During the study period, a total of 446 HCV DAA treatments episodes were provided among 413 individuals. Preliminary analysis of the 413 individuals treated revealed that 37 (9.0%) had evidence of reinfection. Almost all (36/37) cases of reinfection were in people with recent (six month) injecting drug use, so the analysis of other risk factors for reinfection was restricted only to those 36 reinfections in 321 individuals who reported recent injecting drug use. This was because injecting drug use was so clearly an overwhelming risk factor, and analysing factors associated with a single case of reinfection in someone who did not report drug use would not yield interpretable results.

An overview of study participants included in the risk factor and incidence analyses is shown in Figure 1 below.

### 3.1. DAA-Treated Clients with Recent (6 Month) Injecting Drug Use (n = 321)

Demographic, behavioural, and clinical characteristics of all DAA-treated clients (n = 413), those with recent (6 month) injecting drug use (n = 321) and those without recent injecting drug use (n = 92), based on the most recent data available at the time of commencing treatment, are shown in Table 1. Among those with recent (6 month) injecting drug use, the median age was 44 years (IQR = 37–50), a majority were male (71.3%), and about one-quarter (24.9%) were Aboriginal and/or Torres Strait Islander people. A history of incarceration, custody, or juvenile detention was very common (75.1%). About half (46.7%) were receiving opioid agonist therapy (methadone or buprenorphine). Approximately 1 in 11 (9.1%) had been diagnosed with cirrhosis. Results from a sensitivity analysis using client characteristics assessed at the most recent data point (instead of at treatment commencement) were consistent with those shown in Table 1 and were not reported.

The following results are based on the definition of reinfection using all data sources. Among 321 people with recent (6 month) injecting drug use, 285 were not reinfected (i.e., infected once) and 36 (11%) were reinfected (i.e., infected more than once). Among the 36 individuals who were reinfected, 33 were reinfected once (i.e., infected twice) and three were reinfected twice (i.e., infected three times).

We compared the demographic, behavioural, and clinical characteristics (at treatment commencement) of clients not reinfected and reinfected (Table 2). There were notable differences in the proportion reinfected between males (9.2%, 95% CI: 6.1–13.7) and females (16.5%, 95% CI: 9.99–26.0), HIV-positive (20.0%, 95% CI: 7.7–42.9) and HIV-negative (10.6%, 95% CI: 7.6–14.7) people, and HBV-positive (25.0%, 95% CI: 6.3–62.4) and HBV-negative people (11.0%, 95% CI: 8.08–15.0). Only the difference in the proportion reinfected between Aboriginal and/or Torres Strait Islander people (20.0%, 95% CI: 12.6–30.2) and non-Aboriginal and/or Torres Strait Islander people (8.3%, 95% CI: 5.4–12.5) was statistically significant (*p* < 0.01).

The bivariate associations between each characteristic (assessed at treatment commencement) and clients not reinfected and reinfected were investigated (Table 2). In bivariate logistic regression, only Aboriginality was significantly associated with reinfection. Aboriginal and/or Torres Strait Islander origin was associated with an approximate three-fold increase in the odds of HCV reinfection (OR: 2.76, 95% CI: 1.35–5.64, *p* = 0.007).

Aboriginality and gender were statistically significant at the *p* < 0.15 level in the bivariate analysis; however, no other factor met pre-specified criteria for inclusion in the multivariable model. The adjusted associations between Aboriginality and gender with reinfection were calculated. The strength of the associations reduced slightly, and Aboriginality remained significant (Table 3). In the adjusted model, people of Aboriginal and/or Torres Strait Islander origin had odds of reinfection that were 2.73 times greater than people of non-Indigenous origin (OR: 2.73, 95% CI: 1.33–5.60, *p* = 0.006).

### 3.2. DAA-Treated Clients with HCV RNA Interval Testing Data (n = 311)

Among all the individuals treated (n = 413), 102 were lost to follow-up or did not have further HCV testing post-SVR12 date at our service. The remaining 311 clients were included in the analysis of incidence rates, which included 23 reinfections during 246 person years of follow-up, resulting in an overall reinfection rate of 9.5 reinfections/100PY (95% CI: 6.3–14.3).

Sub-group analysis found that reinfection incidence was four times higher in Aboriginal and/or Torres Strait Islander people (20.4/100PY; 95% CI: 12.1–34.4) than non-Aboriginal and/or Torres Strait Islander people (5.1/100PY; 95% CI: 2.7–9.9) (Table 4), and this was the only sub-group where 95% CIs of incidence estimates did not overlap.

There were also substantial differences in other factors. Reinfection incidence was about two times higher in people who were homeless in the past 12 months (13.4/100PY; 95% CI: 8.1–22.2) than those who were not homeless (6.1/10PYy; 95% CI: 3.1–12.3); PWID (past six months; 11.00/100PY; 95% CI: 7.2–16.7) had a reinfection incidence rate that was about five times higher than those who had not injected drugs in the past six months (2.4/100PY; 95% CI: 0.3–17.3). Reinfection incidence was about three times higher in people with a history of incarceration (11.9/100PY; 95% CI: 7.3–19.5) than those without (4.3/100PY; 95% CI: 1.1–17.2). People living with HIV (5.5/100PY; 95% CI: 1.4–22.1) were reinfected (with HCV) at about half the rate of people who were HIV-negative (10.2/100PY: 95% CI: 6.7–15.7). However, the 95% CIs between these groups overlapped.

Of the 311 DAA-treated clients with interval testing data included in the incidence analysis, 237 of these were also people with recent (6 month) injecting drug use and were included in a secondary incidence analysis. The results for reinfection incidence among this subset of clients, overall and by sub-group, followed the same pattern but were marginally higher than the reinfection incidence rates among all DAA-treated clients (Table 4). The overall reinfection rate in people with recent (6 month) injecting drug use was 11.0 reinfections/100PY (95% CI: 7.2–16.7) compared to 9.5 reinfections per 100PY (95% CI: 6.3–14.3) in all treated clients.

We explored whether loss to follow-up, and thus exclusion from incidence calculations, was associated with potential risk factors for HCV reinfection in PWID by comparing those with (n = 237) and without (n = 84) interval testing data and found some risk factors were not evenly distributed. Clients who experienced recent homelessness had about twice the odds of being lost to follow-up (OR: 1.7, 95% CI: 1.02–2.85, *p* = 0.040), whereas being currently on OAT was associated with a 46% decrease in the odds of being lost to follow-up (OR: 0.54, 95% CI: 0.32–0.91, *p* = 0.019).

## 4. Discussion

This study investigated HCV reinfection in socially marginalised clients presenting to an inner-city primary healthcare service in NSW, Australia. To our knowledge, this is the first study of HCV reinfection in this kind of healthcare setting in Australia [23]. The overall proportion of clients ever reinfected was 9%, and the population’s overall incidence rate of HCV reinfection was 9.5/100PY.

Of the demographic, behavioural, and clinical characteristics investigated, only Aboriginality was significantly associated with reinfection. People of Aboriginal and/or Torres Strait Islander origin had odds of HCV reinfection that were almost three times higher and a reinfection incidence rate that was four times higher than people of non-Indigenous background. These data are consistent with a recent Australian study showing that the population reduction in HCV RNA prevalence and increase in treatment uptake was not significant for Aboriginal and Torres Strait Islander people between 2018 and 2021, whereas these markers were significantly improved in non-Indigenous Australians [25], despite evidence of equivalent treatment outcomes if it was provided and follow-up available [26]. Furthermore, Aboriginal and Torres Strait Islanders make up between 20 and 25% of national needle and syringe program attendees, and although their RNA prevalence at such programs is now equivalent to that of non-Indigenous Australians, for several years after DAA availability RNA prevalence rates among Aboriginal attendees did not reduce at the same rate, providing a greater pool of virus to generate new infections over this timeframe [27].

This is unsurprising given that health and social disparities among Aboriginal and/or Torres Strait Islander people are well documented. Disparities include lower levels of education, employment, and income and poorer-quality housing than non-Indigenous Australians, which may lead to higher levels of injection drug use, incarceration, and crime [28]. Institutional racism and intergenerational trauma further contribute to increased HCV reinfection rates, as the literature suggests self-reported racism has strong links with direct psychological harm and increased IDU [29,30]. Our Aboriginal reference group supported these suppositions and identified the practice of cultural sharing of resources (e.g., money, food, accommodation, as well as drugs) among kinship groups and high rates of incarceration as likely contributing factors, as well as recommending caution about people’s comfort identifying as Indigenous in prison or healthcare settings. The group recommended prioritising resourcing for this population and recognising that Aboriginal and Torres Strait Islander communities were diverse, and different messages might be required for different age or gender groups. They emphasised the importance of culturally competent and identifiable Aboriginal and/or Torres Strait Islander staff and peer workers and ensuring the problem was recognised as access to healthcare, testing, harm reduction, and opportunity, rather than Aboriginal and/or Torres Strait Islander culture, per se.

Health resources should therefore be directed towards community health workers, culturally centred care, and community engagement, which are key to achieving the elimination of HCV and closing health disparities between Indigenous and non-Indigenous people [31]. One example is the Deadly Liver Mob program, which is a peer-led intervention which aims to provide HCV screening and treatment to Aboriginal and/or Torres Strait Islander people [32].

In this study, the incidence of HCV reinfection for people who had injected drugs was higher than many comparable prior studies and meta-analyses where the averaged weighted incidence rates ranged between 3.6/100PY and 6.6/100PY [9,13,33,34,35]. However, within these meta-analyses is often a significant heterogeneity of samples, and they did include a small number of studies which reported incidence rates of between 11 and 33% [13]. Several factors may explain the relatively high incidence of HCV reinfection in our study. We used a definition of injecting with a past-6-month timeframe, whereas others used longer cut-offs [36,37] or measures of ongoing injecting drug use [34,38]. We had a very high rate of incarceration and homelessness in our study, and whilst we did not find these to be associated with a significant difference in reinfection risk, they have been identified as risk factors for reinfection in other studies [39,40]. Given the high rate of incarceration, and the lack of needle syringe programs in Australian prisons, this setting is clearly a very important potential driver of reinfection. Furthermore, despite our service operating an integrated needle and syringe program, sub-analyses of surveys from that program indicate that as many as one in five people who inject drugs in our clinic have used a needle and syringe after someone else in the preceding 1 month, providing the opportunity for reinfection to occur [27].

Several factors previously associated with HCV reinfection in other studies, such as OAT provision [13,34], homelessness [39], or incarceration history [40], did not reach statistical significance in our study, although the trends in the odds ratio observed were consistent with expected impact. Despite a higher proportion of people living with HIV being HCV-reinfected, there was also no association between HIV and HCV reinfection rates. This may reflect a longer period of follow-up in this group, as they were prioritised for treatment when DAAs became available, are generally still under care, and are regularly retested.

In fact, numerically, people living with HIV (5.5/100PY; 95% CI: 1.4–22.1) were reinfected (with HCV) at about half the rate of people who were HIV-negative (10.2/100PY: 95% CI: 6.7–15.7). This is in keeping with cohort studies within Australia which report a very low rate of HCV reinfection among people living with HIV [41], where treatment uptake has reduced the RNA prevalence in this population to near-elimination thresholds [42].

In light of the decreasing rate of treatment in Australia, high rates of reinfection of 11.0/100PY in people who currently inject is important in the context of achieving HCV elimination goals by 2030 [1,2]. This suggests the need for interventions which more effectively engage PWID with healthcare services in Australia and aim to prevent reinfection [21,43]. It is currently estimated that roughly one in every six treatments in Australia in 2021 was for reinfection [39].

One intervention currently implemented by our service engages hidden populations of PWID with HCV testing through ‘peer connectors’, who are PWID that identify other people at risk of HCV reinfection without the need to visit a healthcare service. By discovering and recommending our service to these hidden individuals, this raises awareness of HCV testing and treatment options in our local communities. Peer support models more broadly have been shown to increase engagement and can be constructed in different ways [44].

## 5. Limitations

As with all studies, this study had some limitations. There was incomplete follow-up for our population (25% were not followed up with). This could lead to an overestimation of our final incidence calculation if those most at risk of reinfection were more actively followed up with. In contrast, because we relied on data from a single service, reinfections identified and treated in other settings may not appear in our data, underestimating incidence.

We were also unable to capture risk factors at intervals throughout the study, nor collect how they changed around the time of reinfection, and several variables were based on lifetime history reported at initial treatment. Whilst we conducted sensitivity analysis for those variables for which we had updated data and it did not alter the findings, it is still possible that risk factors such as incarceration or homelessness changed during follow-up and influenced reinfection.

Furthermore, there were also very low (n = 1) numbers of reinfections in people not currently injecting drugs, limiting analysis of risk factors in this group.

Our definition of reinfection included clients who were treated twice as part of the reinfected group, which may have led to more reinfections recorded. However, this definition is justified in representing highly marginalised clients who are difficult to follow-up with and commonly retreated in other healthcare facilities such as custodial settings and hospitals that are most representative of PWID.

Study findings are only generalisable to inner-city primary care settings which target marginalised populations, specifically services which engage PWID and Aboriginal and/or Torres Strait Islander people, and may not apply to more rural/regional settings.

## 6. Conclusions

In this clinic population with a focus on people with multiple factors associated with vulnerability, and in whom the majority continued injecting drug use and 70% had a history of incarceration, incidence of reinfection was approximately 9.5% per year and 11% per year in those with recent injecting.

Interventions focusing on maintaining client engagement and follow-up are key to diagnosing and treating HCV reinfection in complex socially marginalised populations and, although challenging to implement, are particularly well suited for delivery in primary health care settings.

Testing, identifying, and retreating reinfection are one part of the solution, but they need to be accompanied by evidence-based harm reduction strategies such as high-coverage needle and syringe programs, low-threshold OAT programs, and access to supervised injecting facilities.

High rates of HCV reinfection in clinic populations with multiple vulnerabilities and continued drug use, especially among Aboriginal and Torres Strait Islander people, highlight the need for ongoing regular HCV testing and retreatment in order to achieve HCV elimination. A priority is resourcing for testing and treatment for Aboriginal and/or Torres Strait Islander people. Our findings support the need for novel and holistic healthcare strategies for PWID and the upscaling of Indigenous cultural approaches and dedicated interventions.

## Figures and Tables

**Figure 1 viruses-16-00957-f001:**
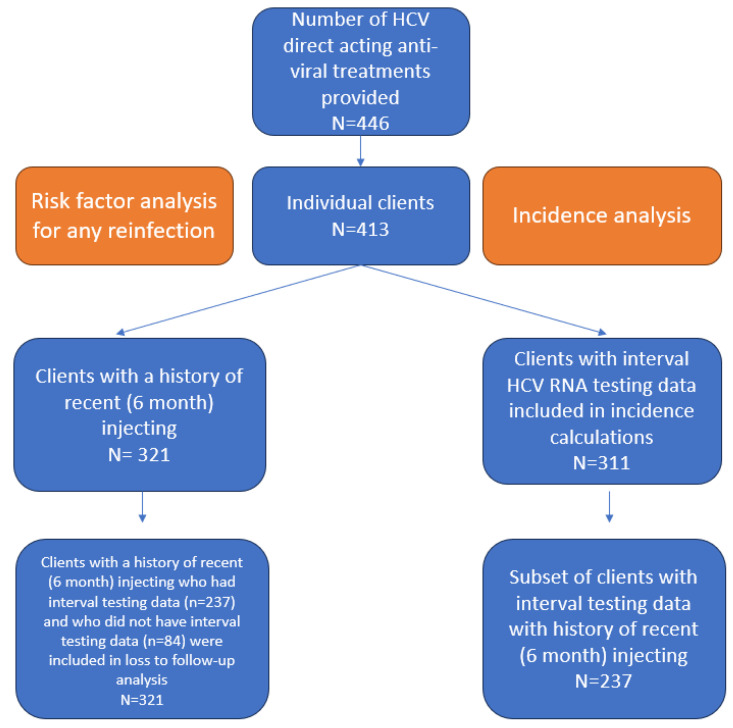
Overview of study participants.

**Table 1 viruses-16-00957-t001:** Demographic, behavioural, and clinical characteristics of all DAA-treated clients, those with recent (6 month) injecting drug use and those without recent injecting drug use, based on the most recent data available at the time of commencing treatment.

Characteristic	All Clients n = 413	Clients without Recent (6 Month) Injecting Drug Use n = 92	Clients with Recent (6 Month) Injecting Drug Use n = 321
	% (n/N)	% (n/N)	% (n/N)
Age (median), in years	45 (IQR = 38–51)	49 (IQR = 42–56)	44 (IQR = 37–50)
Gender			
Male	70.0 (289/413)	65.2 (60/92)	71.3 (229/321)
Female	28.1 (116/413)	33.7 (31/92)	26.5 (85/321)
Non-binary	1.9 (8/413)	1.1 (1/92)	2.2 (7/321)
Aboriginal and/or Torres Strait Islander			
Yes	23.2 (96/413)	17.4 (16/92)	24.9 (80/321)
No	76.8 (317/413)	82.6 (76/92)	75.1 (241/321)
Australian-born			
Yes	84.9 (349/411)	80.4 (74/92)	86.2 (275/319)
No	15.1 (62/411)	19.6 (18/92)	13.8 (44/319)
Men who have sex with men, ever			
Yes	17.0 (70/413)	4.4 (4/92)	20.6 (66/321)
No	83.1 (343/413)	95.7 (88/92)	79.4 (255/321)
Homeless in the last 12 months			
Yes	48.3 (199/412)	30.4 (28/92)	53.4 (171/320)
No	51.7 (213/412)	69.6 (64/92)	46.6 (149/320)
Incarcerated ^1^, ever			
Yes	70.6 (185/262)	52.8 (28/53)	75.1 (157/209)
No	24.2 (77/262)	47.2 (25/53)	24.9 (52/209)
Opioid agonist therapy, current			
Yes	40.7 (168/413)	19.6 (18/92)	46.7 (150/321)
No	59.3 (245/413)	80.4 (74/92)	53.3 (171/321)
HIV-positive ^2^			
Yes	5.6 (23/413)	3.3 (3/92)	6.2 (20/321)
No	94.4 (390/413)	96.7 (89/92)	93.8 (301/321)
HBV-positive ^2^			
Yes	2.0 (8/408)	-	2.5 (8/316)
No	98.0 (400/408)	100.0 (92/92)	97.5 (308/316)
Cirrhotic ^3^			
Yes	10.7 (44/412)	16.3 (15/92)	9.1 (29/320)
No	89.3 (368/412)	83.7 (77/92)	90.9 (291/320)

^1^ Including ever in custody or juvenile detention; ^2^ at start of HCV treatment; ^3^ Fibroscan > 12.5 kpa; OR = odds ratio; 95% CI = 95% confidence interval; HIV = human immunological virus; HBV = hepatitis B virus; IQR = interquartile range.

**Table 2 viruses-16-00957-t002:** Table of (1) demographic, behavioural, and clinical characteristics at treatment commencement of clients with recent (6 month) injecting drug use not reinfected and reinfected and (2) bivariate associations between each characteristic and reinfection.

Characteristic	(1) Clients with Recent (6 Month) Injecting Drug Use					(2) Bivariate Association		
	Not Reinfected		Reinfected					
	% (n/N)	95% CI	% (n/N)	95% CI	*p*-Value	OR	95% CI	*p*-Value
Mean age, in years	43.8	42.8–45.0	41.7	38.5–45.0	0.199	0.98	0.94–1.01	0.199
Gender					0.184			
Male	90.8 (208/229)	86.3–94.0	9.2 (21/229)	6.0–13.7		1	-	-
Female	83.5 (71/85)	74.0–90.2	16.5 (14/85)	10.0–26.0		1.95	0.94–4.04	0.071
Non-binary	85.7 (6/7)	41.7–98.1	14.3 (1/7)	2.0–58.3		1.65	0.19–14.37	0.650
Aboriginal and/or Torres Strait Islander					0.004 *			
No	91.7 (221/241)	87.5–94.6	8.3 (20/241)	5.4–12.5		1	-	-
Yes	80.0 (64/80)	69.8–87.3	20.0 (16/80)	12.6–30.2		2.76	1.35–5.64	0.007 *
Australian-born					0.313			
No	93.2 (41/44)	80.8–97.8	6.8 (3/44)	2.2–19.2		1	-	-
Yes	88.0 (242/275)	83.6–91.3	12.0 (33/275)	8.7–16.4		1.86	0.54–6.36	0.285
Men who have sex with men, ever					0.484			
No	89.4 (228/255)	85.0–92.7	10.6 (27/255)	7.4–15.0		1	-	-
Yes	86.4 (57/66)	75.8–92.8	13.6 (9/66)	7.2–24.2		1.33	0.59–2.99	0.493
Homeless in the last 12 months					0.327			
No	90.6 (135/149)	84.7–94.4	9.4 (14/149)	5.63–15.3		1	-	-
Yes	87.1 (149/171)	81.2–91.4	12.9 (22/171)	8.61–18.8		1.42	0.70–2.89	0.324
Incarcerated ^1^, ever					0.372			
No	88.5 (46/52)	76.5–94.8	11.5 (6/52)	5.3–23.5		1	-	-
Yes	86.6 (136/157)	80.3–91.1	13.4 (21/157)	8.9–19.7		1.18	0.45–3.11	0.732
Opioid agonist therapy, current					0.771			
No	88.3 (151/171)	82.5–92.3	11.7 (20/171)	7.7–17.5		1	-	-
Yes	89.3 (134/150)	83.3–93.4	10.7 (16/150)	6.6–16.7		0.90	0.45–1.81	0.770
HIV-positive ^2^					0.199			
No	89.4 (269/301)	85.3–92.4	10.6 (32/301)	7.6–14.7		1	-	-
Yes	80.0 (16/20)	57.1–92.3	20.0 (4/20)	7.7–42.9		2.10	0.66–6.67	0.237
HBV-positive ^2^					0.220			
No	89.0 (274/308)	84.9–92.0	11.0 (34/308)	7.98–15.07		1	-	-
Yes	75.0 (6/8)	37.6–93.7	25.0 (2/8)	6.3–62.4		2.7	0.52–13.84	0.277
Cirrhotic ^3^					0.871			
No	89.0 (258/291)	84.5–91.8	11.3 (33/291)	8.2–15.5		1	-	-
Yes	90.0 (26/29)	72.3–96.6	10.3 (3/29)	3.4–27.7		0.90	0.26–3.14	0.870

^1^ Including ever in custody or juvenile detention; ^2^ at start of HCV treatment; ^3^ Fibroscan > 12.5 kpa; * *p* < 0.01; OR = odds ratio; 95% CI = 95% confidence interval; sd = standard deviation; HIV = human immunological virus; HBV = hepatitis B virus.

**Table 3 viruses-16-00957-t003:** Multivariable association between reinfection, Aboriginality, and gender among clients who inject drugs (past 6 months; N = 321).

Characteristic	Adjusted Association		
	AOR	95% CI	*p*-Value
Aboriginal and/or Torres Strait Islander ^1^	2.73	1.33–5.60	0.006 *
Gender ^2^			
Female	1.92	0.92–4.02	0.084
Non-binary	1.59	0.18–14.28	0.681

^1^ Reference group = No; ^2^ reference group = male; * *p* < 0.05; AOR = adjusted odds ratio; 95% CI = 95% confidence interval.

**Table 4 viruses-16-00957-t004:** Reinfection rate per 100 person years of (1) DAA-treated clients with HCV RNA interval testing data and (2) a subset of clients with recent (6 month) injecting drug use.

Characteristic ^2^	DAA-Treated Clients with HCV RNA Interval Testing Data ^1^		Subset of Clients with Recent (6 Month) Injecting Drug Use	
	Reinfection Rate/100PY	95% CI	Reinfection Rate/100PY	95% CI
Overall	9.5	6.3–14.3	11.00	7.2–16.7
Gender				
Male	10.047	6.2–16.4	11.9	7.3–19.5
Female	9.2	4.4–19.2	9.9	4.5–22.1
Non-binary	- ^3^	-	-	-
Aboriginal and/or Torres Strait Islander				
Yes	20.4	12.1–34.4	20.5	11.9–35.4
No	5.1	2.7–9.9	6.4	3.4–12.4
Australian-born				
Yes	10.2	6.6–15.6	11.6	7.5–17.9
No	5.8	1.5–23.3	7.3	1.8–29.1
Men who have sex with men, ever				
Yes	9.5	4.5–19.9	10.3	4.9–21.5
No	9.9	5.1–19.0	12.6	6.5–24.2
Homeless in the last 12 months				
Yes	13.4	8.1–22.2	15.5	9.4–25.7
No	6.1	3.1–12.2	6.7	3.2–14.0
Incarceration ^4^, ever				
Yes	11.9	7.3–19.5	13.9	8.5–22.7
No	4.3	1.1–17.2	2.8	0.4–19.6
Recent (6 months) injecting drug use				
Yes	11.0	7.2–16.7	-	-
No	2.4	0.3–17.3	-	-
Opioid agonist therapy, current				
Yes	10.5	5.9–18.4	10.2	5.7–18.4
No	8.6	4.8–15.6	11.8	6.5–21.3
HIV-positive ^5^				
Yes	5.5	1.4–22.1	6.8	1.7–27.1
No	10.2	6.7–15.7	11.7	7.6–18.2
HBV-positive ^5^				
Yes	9.5	1.3–67.4	9.5	1.3–67.4
No	9.7	6.4–14.7	11.3	7.4–17.4
Cirrhotic ^6^				
Yes	14.6	6.1–35.0	18.2	7.6–43.6
No	8.8	5.6–14.00	10.0	6.2–16.1

^1^ Including people who did not report past six months injecting drug use; ^2^ reinfection rate not computed for age as continuous variable; ^3^ could not compute due to low sample size; ^4^ including ever in custody or juvenile detention; ^5^ at start of HCV treatment; ^6^ Fibroscan > 12.5 kpa; OR = odds ratio; 95% CI = 95% confidence interval; HIV = human immunological virus; HBV = hepatitis B virus; py = person years.

## Data Availability

As these data are drawn from clinic-based databases, they are not publicly available.

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
