# Peer review of "Hepatitis C (HCV) Reinfection and Risk Factors among Clients of a Low-Threshold Primary Healthcare Service for People Who Inject Drugs in Sydney, Australia"

_viruses, 2024, doi:10.3390/v16060957_

Round 1

Reviewer 1 Report

Comments and Suggestions for Authors

I have with great interest read the manuscript "Hepatitis C (HCV) Reinfection and Risk Factors among Clients of a Low Threshold Primary Healthcare Service for People Who Inject Drugs in Sydney, Australia" by P. Read et al. 

The study focus on HCV reinfection among PWID in a primary health care setting, targeting socially marginalised clients in NSW, Australia. Overall reinfection rate was 9.5/110 PY. Incidence rates varied by sub-group and were highest for Aboriginal and/or Torres Strait Islander people which is thoroughly discussed in the manuscript. 

I find this manuscipt interesting, and adding to the growing data on HCV reinfection among PWID in different settings. However, I believe that the manuscript could be improved with some clarifications and adjustments. 

Major

L 32: Please check statement of ‘reduced HCV prevalence by 90%’ and reference. Do you mean incidence? Or 80% reduction of prevalence, check page 40? Or incidence 80%, page xiii?

“In the Global Health Sector Strategy (GHSS) for viral hepatitis, elimination of viral hepatitis as a public health problem by 2030 is defined as a 90% reduction in incidence (95% for HBV and 80% for HCV) and a 65% reduction in mortality from HBV and HCV, compared with the 2015 baseline (4)"

4. Global Health Sector Strategy on viral hepatitis 2016–2021. Geneva: World Health Organzation; 2016 (https://www.who.int/hepatitis/strategy2016-2021/ghss-hep/en/)”

L51: What is suggested by “..comparable following interferon-based and DAA therapy”?

L100: How is lost to follow-up ‘defined’ in the database?

L103-105: Why is 2.4 separated to 2.5?

L106: Based on most recent data or last test date (L112)?

L123-151: The defined classifications are quite complex - and hard to read. Consider rewriting and take inspiration from other papers describing this methodology (actuarial method)

L123L130: Why are individuals not tested for reinfection classified as not reinfected? Why are they not classified as lost-to-follow-up (LTFU)? Compare rationale in L149-151

Statistical analysis: I suggest that the statistical analysis section is rewritten to be more condensed/specific for smoother reading.

L192: The rational for not including all 37 cases  is not clear to me, as recent IDU (n=36) is self reported data

Table 1: If you want to compare client with recent IDU and those without recent IDU, isn’t better to divide the groups that way, rather than comparing those with recent IDU with all (since the majority had recent IDU)? Consider presenting: all, non-recent IDU, recent IDU.

Table 1: Be consistent regarding one or two decimals

L: 215-219: The idea of reporting the total number of infections is not clear to me, as participants may hav been infected several times before, but spontaneously cleared or treated.

Table 2: Same as in table 1. If you want to compare client with recent IDU and those without recent IDU, isn’t better to divide the groups that way, rather than comparing those with recent IDU with all? Consider presenting: all, non-recent IDU, recent IDU.

L329-334: There are several international PWID reinfection studies showing reinfection rates around 10/100 PY and above, which should be mentioned/referred to

L337-341: Is there specific study unit (KRC) data in reference number 40?

Minor

L36: ‘These medications’. Unclear, although I understand that you refer to DAAs.

L40: spontaneous clearance, rather than natural clearance?

L44: MSM = men who have sex with men, as referred to later in L113, not male-to-male sex

L48: Easier to read if abbreviated to 6.2/100 PY, and throughout in the manuscript

L66: PWID already explained in L45

L224: ‘people’ after the parentheses

Incidence figures: Does two decimals in incidence figures add anything compared to report only one decimal?

Author Response

Dear Viruses Editorial Committee

Re: Hepatitis C (HCV) reinfection and risk factors among clients of a low threshold  primary healthcare service for people who inject drugs in Sydney, Australia

Thank you for the opportunity to revise our paper, and for the helpful comments from the reviewers.

We have been able to address almost all comments, and detail the specific queries and responses underneath. In the few cases we have not made the suggested change, we detail our rationale.

We are happy to revisit these points should the editor request this.

We look forward to hearing from you, and attach a mark up and clean copy in our resubmission

Reviewer 1

Major

L 32: Please check statement of ‘reduced HCV prevalence by 90%’ and reference. Do you mean incidence? Or 80% reduction of prevalence, check page 40? Or incidence 80%, page xiii?

We have checked the reference, and the reviewer is correct and we have made these changes to reflect incidence, and that the HCV component is a reduction of 80%.

L51: What is suggested by “..comparable following interferon-based and DAA therapy”?

Thank you. The paper finds that there was no significant difference between reinfection rates in the interferon-based groups and the DAA based groups. We have amended the manuscript to make this clearer.

L100: How is lost to follow-up ‘defined’ in the database?

We have clarified that LFTU was defined as no further RNA testing after treatment initiation

L103-105: Why is 2.4 separated to 2.5?

I am afraid I do not know what this comment relates to. We are happy to make a change if clarified.

L106: Based on most recent data or last test date (L112)?

We describe in the text that this is the data available at time of treatment.

L123-151: The defined classifications are quite complex - and hard to read. Consider rewriting and take inspiration from other papers describing this methodology (actuarial method)

We have amended the methods to take this advice into account.

L123L130: Why are individuals not tested for reinfection classified as not reinfected? Why are they not classified as lost-to-follow-up (LTFU)? Compare rationale in L149-151

This comment only relates to the analysis of the proportion diagnosed with reinfection. We recognize people may be reinfected elsewhere, or have tested negative elsewhere and not known to us. Therefore the only certain variable was whether they had reinfected, not whether they had not. Thus we were interested in the probability of being diagnosed with reinfection, not the probability of being infection free as this was difficult to prove. Therefore we included all people we would have wished to test, regardless of whether they received that test. We have clarified the text to make it clear we were analyzing the proportion diagnosed with reinfection.

With respect to incidence data, this only included those with confirmed negative and/or positive RNA test dates. Therefore this analysis did not include LFTU individuals, as they had no data to contribute to time under follow-up.

Statistical analysis: I suggest that the statistical analysis section is rewritten to be more condensed/specific for smoother reading.

We have condensed the statistical analysis section

L192: The rational for not including all 37 cases  is not clear to me, as recent IDU (n=36) is self reported data

We received informal statistical advice that attempting to analyse risk factors for reinfection among people who did not report ongoing drug use was futile given the cell size was n=1. Risk factors would end up being a description of that individual person, which would also not have provided sufficient confidentiality or interpretation. We have clarified this is the text.

Table 1: If you want to compare client with recent IDU and those without recent IDU, isn’t better to divide the groups that way, rather than comparing those with recent IDU with all (since the majority had recent IDU)? Consider presenting: all, non-recent IDU, recent IDU.

As per above answer with respect to over-analysing  a single case of reinfection among someone who did not report drug use. We do have this data if the editors feel it is useful to include. However, we have included demographic data for non-recent IDU in table 1 as suggested.

Table 1: Be consistent regarding one or two decimals

We have ensured that all data is to 1 decimal point in table 1

L: 215-219: The idea of reporting the total number of infections is not clear to me, as participants may have been infected several times before, but spontaneously cleared or treated.

This is just descriptive data and shows that some people are diagnosed multiple times, so we have not deleted it. We can do if the editors prefer.

Table 2: Same as in table 1. If you want to compare client with recent IDU and those without recent IDU, isn’t better to divide the groups that way, rather than comparing those with recent IDU with all? Consider presenting: all, non-recent IDU, recent IDU.

As per above answer with respect to over-analysing  a single case of reinfection among someone who did not report drug use. We do have this data if the editors feel it is useful to include. If we present risk factors for reinfection among non-IDU when there is only 1 case, then it will simply reflect that particular person’s characteristics and not be hugely meaningful.

L329-334: There are several international PWID reinfection studies showing reinfection rates around 10/100 PY and above, which should be mentioned/referred to

Thank you, rather than adding more references, we have amended the discussion to note that within the referenced metanalyses, there were studies that reported an incidence rate of >10%.

L337-341: Is there specific study unit (KRC) data in reference number 40?

This is based on unpublished sub-analysis of this report. We have added a comment to clarify this.

Minor

L36: ‘These medications’. Unclear, although I understand that you refer to DAAs.

We have added DAA to clarify

L40: spontaneous clearance, rather than natural clearance?

We have changed to spontaneous clearance

L44: MSM = men who have sex with men, as referred to later in L113, not male-to-male sex

We have made these two sections consistent.

L48: Easier to read if abbreviated to 6.2/100 PY, and throughout in the manuscript

We have written person/years the first time it is used, then abbreviated to PY thereafter as suggested

L66: PWID already explained in L45

Amended

L224: ‘people’ after the parentheses

Apologies, I am not sure what this is referring to. I am happy to make changes if required.

Incidence figures: Does two decimals in incidence figures add anything compared to report only one decimal?

We have amended incidence data and the 95%CI around proportions to 1 decimal point as suggested. We have left the ORs with 2 decimal points as they are much smaller numbers and describing by 2 decimal points shows small differences between the groups

Reviewer 2 Report

Comments and Suggestions for Authors

Overall

It has been a pleasure to review “Hepatitis C (HCV) Reinfection and Risk Factors among Clients of a Low Threshold Primary Healthcare Service for People Who Inject Drugs in Sydney, Australia”. This manuscript presents findings from a retrospective cohort of individuals being treated for HCV in routine clinical primary care. Authors provide a concise report with recognition of limitations. The majority of my concerns are regarding the methods and data analysis. Below are my recommendations by section.

Introduction

The authors did a good job at providing clarity on the importance of examining HCV reinfection in Australia. I only have a few comments:

1.    One thing that I felt was missing from the introduction was the role that needle and syringe programs play in the prevention of HCV reinfection. Authors highlight the need for additional treatment allocation to folks that are reinfected, but some commentary on client engagement in NSP, given that the majority of reinfections will be PWID, is warranted.

2.    Second paragraph could be extended to highlight that DAAs are curative in nature with high rates of SVR. I imagine the readers will understand the HCV landscape but this is useful information to present.

3.    Authors should define what mainstream healthcare services are since this paper is looking at a primary healthcare service (how is your service different then mainstream healthcare in Australia?)

Methods

This section of the manuscript lacks detail. I have a few comments:

1.    Does KRC provided any harm reduction services alongside their primary healthcare service?

2.    Was this study using retrospective data or was it truly prospective in nature? If prospective, were clients aware of their involvement in the research? Based on the IRB/ethics statement looks like these data would be considered retrospective.

3.    I think the authors could provide more detail regarding how this dataset was created and from what data sources. As currently written, it is very unclear how the dataset was constructed through the various retrospective/prospective components.

4.    Providing a bullet list of variables that were included seems like an inappropriate way of presenting these data for publication purposes. I would suggest the authors use traditional editorial techniques to provide this information.

5.    Models would be considered bivariate and multivariable, not multivariate. These mean two different things and the way the statistical analysis is presented suggested a multivariable logistic regression model was used.

6.    Potential bias needs to be addressed/discussed regarding grouping folks who did not test for reinfection as not reinfected.

7.    Authors need to provide a justification as to way variables only at p<0.05 on bivariate analyses were included in the multivariable logistic regression. Previous studies have shown that inclusion of only 0.05 can fail to identify variables known to be important. Current recommendations are 0.15. This needs to be considered and models should be re-estimated accordingly. 

Results

Results are adequately presented given the data analysis presented in the methods section.

Discussion

The discussion is well written and cohesively discusses the findings of the study. No further comments.

Author Response

20-5-24

Dear Viruses Editorial Committee

Re: Hepatitis C (HCV) reinfection and risk factors among clients of a low threshold  primary healthcare service for people who inject drugs in Sydney, Australia

Thank you for the opportunity to revise our paper, and for the helpful comments from the reviewers.

We have been able to address almost all comments, and detail the specific queries and responses underneath. In the few cases we have not made the suggested change, we detail our rationale.

We are happy to revisit these points should the editor request this.

We look forward to hearing from you, and attach a mark up and clean copy in our resubmission

Reviewer 2

Introduction

  1. One thing that I felt was missing from the introduction was the role that needle and syringe programs play in the prevention of HCV reinfection. Authors highlight the need for additional treatment allocation to folks that are reinfected, but some commentary on client engagement in NSP, given that the majority of reinfections will be PWID, is warranted.

Have added a sentence about the importance of NSP and OAT in reinfection prevention

  1. Second paragraph could be extended to highlight that DAAs are curative in nature with high rates of SVR. I imagine the readers will understand the HCV landscape but this is useful information to present.

Have added the high cure rates to the paper

  1. Authors should define what mainstream healthcare services are since this paper is looking at a primary healthcare service (how is your service different then mainstream healthcare in Australia?)

We have added extra content to describe KRC further, and what we mean by mainstream healthcare 

Methods

  1. Does KRC provided any harm reduction services alongside their primary healthcare service?

Yes. We have detailed some of the harm reduction services available in the text.

  1. Was this study using retrospective data or was it truly prospective in nature? If prospective, were clients aware of their involvement in the research? Based on the IRB/ethics statement looks like these data would be considered retrospective.

We have clarified that the overall dataset was generated retrospectively from a number of sources

  1. I think the authors could provide more detail regarding how this dataset was created and from what data sources. As currently written, it is very unclear how the dataset was constructed through the various retrospective/prospective components.

As above

  1. Providing a bullet list of variables that were included seems like an inappropriate way of presenting these data for publication purposes. I would suggest the authors use traditional editorial techniques to provide this information.

We have changed this section to continuous prose as suggested.

  1. Models would be considered bivariate and multivariable, not multivariate. These mean two different things and the way the statistical analysis is presented suggested a multivariable logistic regression model was used.

We have clarified this terminology  where relevant in the paper.

  1. Potential bias needs to be addressed/discussed regarding grouping folks who did not test for reinfection as not reinfected.

We believe this point is already addressed in the limitations section of the discussion

  1. Authors need to provide a justification as to way variables only at p<0.05 on bivariate analyses were included in the multivariable logistic regression. Previous studies have shown that inclusion of only 0.05 can fail to identify variables known to be important. Current recommendations are 0.15. This needs to be considered and models should be re-estimated accordingly. 

 The model has been re-estimated in line with reviewer’s comments. We note this has not changed the variables included in the model or the results.

Reviewer 3 Report

Comments and Suggestions for Authors

This is a well written analysis of high importance regarding risk factors for HCV-reinfection in an understudied population in Australia. There are some minor revisions requested for clarity in the methods as well as rounding out the discussion.

1) Please provide more clarity around definitions and rationale for both classifications of HCV re-infection. For definition 1, the first sentence seems to imply that all individuals considered in this cohort have an HCV RNA at some point but I don't believe that this is true given the second sentence. Would plainly state that this cohort included all individuals whether or not they had a repeat HCV RNA (if this is true). An example of other data sources (outside of RNA) might be helpful for definition 1, e.g. clinical note document, prescription of another DAA regimen, etc. 

2) In the discussion, please expound upon "cultural sharing practices among kinship groups". Does this refer to practices such as equipment use that might increase risk, or other practices that might be protective?

3) Discussion around harm reduction services e.g. sterile equipment services is limited, and seems to be a key factor in preventing reinfection. Are there additional data that can contextualize the existence or lack of services provided for Aboriginal and/or Torres Strait Islander communities, as well as the community at large? The authors mention their program several times but do not provide further comment on uptake or access. More context in the discussion is warranted. 

4) While homelessness and incarceration did not reach significance as factors for re-infection, the trend was observable in the data (in regards to line 334-337 on page 10). 

5) typo: "Australis" on line 359 page 11

6) Given the very high prevalence of incarceration in the total sample, would include more in the discussion in regards to potential interventions in the criminal legal system in Australia, for example, are OAT programs widespread or limited in this setting? Are there there peer navigation programs linking people to treatment at re-entry?

7) In the limitations, the incomplete follow up and imbalance regarding homelessness could lead to underestimation (as opposed to risk of overestimation as suggested by authors) of re-infection, particularly since this has been shown to be a significant risk factor in other studies. 

Author Response

20-5-24

Dear Viruses Editorial Committee

Re: Hepatitis C (HCV) reinfection and risk factors among clients of a low threshold  primary healthcare service for people who inject drugs in Sydney, Australia

Thank you for the opportunity to revise our paper, and for the helpful comments from the reviewers.

We have been able to address almost all comments, and detail the specific queries and responses underneath. In the few cases we have not made the suggested change, we detail our rationale.

We are happy to revisit these points should the editor request this.

We look forward to hearing from you, and attach a mark up and clean copy in our resubmission

Reviewer 3

  1. Please provide more clarity around definitions and rationale for both classifications of HCV re-infection. For definition 1, the first sentence seems to imply that all individuals considered in this cohort have an HCV RNA at some point but I don't believe that this is true given the second sentence. Would plainly state that this cohort included all individuals whether or not they had a repeat HCV RNA (if this is true). An example of other data sources (outside of RNA) might be helpful for definition 1, e.g. clinical note document, prescription of another DAA regimen, etc. 

Thank you. In relation to this, and other reviewer comments on this section we have amended to provide further clarity.

  1. In the discussion, please expound upon "cultural sharing practices among kinship groups". Does this refer to practices such as equipment use that might increase risk, or other practices that might be protective?

This refers to a cultural expectation of sharing with close family and friends, including money, resources, transport, food etc, it does not specifically relate to drug paraphernalia sharing. We have clarified this in the text.

  1. Discussion around harm reduction services e.g. sterile equipment services is limited, and seems to be a key factor in preventing reinfection. Are there additional data that can contextualize the existence or lack of services provided for Aboriginal and/or Torres Strait Islander communities, as well as the community at large? The authors mention their program several times but do not provide further comment on uptake or access. More context in the discussion is warranted. 

We have provided more data in the discussion on needle syringe program access, including for Aboriginal people.

  1. While homelessness and incarceration did not reach significance as factors for re-infection, the trend was observable in the data (in regards to line 334-337 on page 10). 

Yes we agree, and have mentioned this in the discussion

  1. typo: "Australis" on line 359 page 11

Corrected. Thank you.

  1. Given the very high prevalence of incarceration in the total sample, would include more in the discussion in regards to potential interventions in the criminal legal system in Australia, for example, are OAT programs widespread or limited in this setting? Are there peer navigation programs linking people to treatment at re-entry?

We have added some commentary about the high lifetime incarceration rate for this population. However, a detailed exploration about the contribution of incarceration to reinfection, and the range of interventions (or otherwise) available in the justice system is probably beyond the scope of this paper.

  1. In the limitations, the incomplete follow up and imbalance regarding homelessness could lead to underestimation (as opposed to risk of overestimation as suggested by authors) of re-infection, particularly since this has been shown to be a significant risk factor in other studies. 

Yes this is possible. We discuss the potential for over or under estimation based on Loss to follow up in the discussion.

Round 2

Reviewer 2 Report

Comments and Suggestions for Authors

All comments have been addressed

Comments on the Quality of English Language

N/A